# Lithiated porous silicon nanowires stimulate periodontal regeneration

Martti Kaasalainen [1,17], Ran Zhang[2,17], Priya Vashisth[1], Anahid Ahmadi Birjandi[1], Mark S'Ari[3], Davide Alessandro Martella [1], Mark Isaacs[4,5], Ermei Mäkilä[6], Cong Wang [1], Evelin Moldenhauer[7], Paul Clarke[7], Alessandra Pinna[8,9,10], Xuechen Zhang [1], Salman A. Mustfa[1], Valeria Caprettini[1], Alexander P. Morrell [11], Eileen Gentleman [1], Delia S. Brauer[12], Owen Addison [11], Xuehui Zhang [13], Mads Bergholt[1], Khuloud Al-Jamal [14], Ana Angelova Volponi[1], Jarno Salonen [6], Nicole Hondow [3], Paul Sharpe [1,15] & Ciro Chiappini [1,16] ✉

Periodontal disease is a significant burden for oral health, causing progressive and irreversible damage to the support structure of the tooth. This complex structure, the periodontium, is composed of interconnected soft and mineralised tissues, posing a challenge for regenerative approaches. Materials combining silicon and lithium are widely studied in periodontal regeneration, as they stimulate bone repair via silicic acid release while providing regenerative stimuli through lithium activation of the Wnt/β-catenin pathway. Yet, existing materials for combined lithium and silicon release have limited control over ion release amounts and kinetics. Porous silicon can provide controlled silicic acid release, inducing osteogenesis to support bone regeneration. Prelithiation, a strategy developed for battery technology, can introduce large, controllable amounts of lithium within porous silicon, but yields a highly reactive material, unsuitable for biomedicine. This work debuts a strategy to lithiate porous silicon nanowires (LipSiNs) which generates a biocompatible and bioresorbable material. LipSiNs incorporate lithium to between 1% and 40% of silicon content, releasing lithium and silicic acid in a tailorable fashion from days to weeks. LipSiNs combine osteogenic, cementogenic and Wnt/β-catenin stimuli to regenerate bone, cementum and periodontal ligament fibres in a murine periodontal defect.

Periodontal disease is the most prevalent oral disease. WHO's latest reports indicate that around 19% of the global adult population is suffering from its severe form, characterised by irreversible destruction of the periodontium, causing tooth loss[1]. The periodontium is a complex structure supporting the tooth, composed of the gingiva (soft tissue) protecting and sealing the cervical part of the tooth within the oral cavity; the periodontal ligament (soft tissue) that anchors the tooth through the cementum (mineralised tissue) to the alveolar bone (mineralised tissue). Regenerating such complex arrangement of mineralised and soft tissues, and their interfaces, is a longstanding challenge for tissue engineering, making the periodontium a good model to benchmark regenerative strategies[2,3]. While periodontal regenerative approaches are becoming available in the clinic and provide some degree of alveolar bone restoration, they have limited effect on the other tissues involved[4].

Silicon-based materials are osteoinductive and play an important role in mineralised tissue regeneration, including for the treatment of periodontitis[5]. Incorporating lithium within silicon-based biomaterials is particularly appealing to strengthen their regenerative capacity. Lithium is a potent GSK3 inhibitor, stimulating Wnt/β-catenin signalling[6,7]. The Wnt/β-catenin pathway is a master regulator of osteogenesis, cell differentiation, and proliferation, providing established regenerative stimuli for mineralised[8] and soft tissues[9,10]. Lithium can stimulate bone formation in mice[11] and decrease bone fracture risk in humans[12]. Wnt/β-catenin signalling promotes proliferation and differentiation of human periodontal ligament cells (hPDLCs)[13], and lithium ions promote their cementogenic differentiation, inducing regeneration of cementum and periodontal ligament fibres[7,14].

Bioactive glasses have established the crucial role of silicic acid and lithium ions in bone regeneration, encountering widespread clinical success[15]. Aluminosilicate clays containing lithium, such as laponite, can also provide sustained release of lithium and silicon ions and induce osteogenesis[16]. Yet, constraints in material formulation and control over release kinetics limit the contribution that lithium can bring to the regenerative efficacy of these approaches[17–19].

Porous silicon is a compelling alternative to improve ion release kinetics for tissue regeneration. Porous silicon has an established osteogenic potential[20], high biocompatibility[21,22], tuneable particle size[23–25] and tailorable bioresorption[26]. The dissolution of porous silicon regulates ion release kinetic, which can be tailored from a few hours to days simply by controlling its porous structure during formation[27]. Further surface derivatisation can extend release over several months[28]. Strategies for doping silicon with high precision using several elements across the periodic table are highly established in the semiconductor industry. These strategies are directly applicable to porous silicon[29], highlighting a path to exploit the dissolution of porous silicon for the controlled release of its dopants. In particular, porous silicon can theoretically host up to 52 wt% of lithium ($Li_{22}Si_5$)[30], while providing a platform for controlled ion release, through its tailorable dissolution kinetics. Several prelithiation strategies have been developed in lithium battery technology to incorporate large amounts of lithium within the crystal structure of silicon[31–33]. Particularly, the prelithiation of porous silicon nanowires has been extensively investigated, since the porous structure assists maintaining anode integrity through the volumetric expansions and contractions arising from lithium-ion exchange during battery cycling[29,34,35]. While pre-lithiation allows incorporating large and controllable amounts of lithium within porous silicon, it typically yields pyrophoric and highly reactive materials[36]. It would be appealing to develop similarly effective lithiation approaches suitable for biological applications.

## Results

### Porous silicon lithiation

The potential of lithiated porous silicon as a competitive regenerative material for combined lithium and silicon release led us to develop a lithiation approach to generate biocompatible, bioresorbable LipSiNs. The parameters used to optimise the lithiation process are summarised in Supplementary Fig. 1. This prelithiation strategy uses porous silicon nanowires (pSiNs) generated via metal assisted chemical etching[37] (MACE, Fig. 1a) with controllable porosity ranging between 51 % and 63 % (Supplementary Fig. 2). The pSiNs are mixed with a lithium precursor such as LiOH and LiCl. Mixing can occur in the solid state by grinding pSiNs with the lithium precursor in a mortar, or in the liquid phase by adding pSiNs to aqueous or methanolic lithium solutions (Fig. 1b, Supplementary Fig. 3). Heating this mixture induces lithiation, producing LipSiNs (Fig. 2a, b). The lithiation process occurs in the presence of oxygen and/or nitrogen as well as humidity. The exposure of the particles to oxidising and/or nitriding agents yields LipSiNs which are inert when exposed to air and biological fluids. The lithium precursor, the ratio of Li:Si precursors, atmosphere, temperature, and time are key parameters determining the amount of lithium incorporation (Fig. 1c, Supplementary Fig. 4). The Li/Si mass ratio ranges from 1 % for LiOH at 450 °C, up to 40 % for LiCl at 800 °C (Fig. 1c), providing a broad range of accessible lithium content.

### Physicochemical characterisation

Lithiation preserves the shape of nanowires (Fig. 2a, b) which retain their length and diameter (Supplementary Fig. 6). LipSiNs remain mesoporous, although with reduced specific surface areas (Fig. 2c, Supplementary Figs. 2 and 5) as expected from the volumetric expansion induced by lithium insertion[38]. Centrifugal field-flow fractionation (CF3) combined with in-line dynamic light scattering (DLS), multiangle light scattering (MALS), and inductively coupled plasma mass spectrometry (ICPMS), abbreviated as CF3-MALS-ICPMS, was used to determine the size distribution and elemental composition of LipSiNs in comparison to the precursor pSiNs. LipSiN containing 5% lithium (LipSiN-5%) and pSiNs show similar nanowire concentrations across all size fractions (ICPMS, Fig. 2d) and similar distributions for radius of gyration (MALS) and hydrodynamic radius (DLS) (Fig. 2e).

High resolution transmission electron microscopy (HRTEM) of LipSi-5% show the nanowires extending along the <100> direction with an inner core single-crystal structure surrounded by an amorphous shell (Fig. 3a). LipSi 1.2% show a similar structure, with a thinner shell. LipSi-4% instead remains crystalline throughout, analogously to pSiNs. The processing temperature of 650 °C, above the melting point of lithium chloride likely improves the doping and annealing process, contributing to the fully crystalline structure of LipSi 4%, unlike the 450 °C temperature, below LiCl melting point, used for LipSi-5% and LipSi-1.2%. Electron energy loss spectroscopy (EELS) of LipSi-5% show the presence of silicon and lithium thorough both the amorphous and crystalline phases. Furthermore, the amorphous shell displays a higher Li/Si ratio with respect to the core (Fig. 3b, Supplementary Fig. 7). The relative intensity of peaks within the fine structure of the silicon absorption edge varies between the shell and the core, suggesting a different coordination across the two regions. The elemental analysis of the size-fractionated nanowire population (CF3-MALS-ICPMS) confirms the simultaneous presence of lithium and silicon throughout all fractions (Fig. 2d).

X-ray photoelectron spectroscopy (XPS) analysis of the core level spectra of Si 2p and Li 1s before and after thermal treatment at 450 °C confirms the incorporation of lithium within the silicon nanowires (Supplementary Fig. 8). The Si 2p spectra show increasing oxidation between the lithium and silicon mixture and the treated LipSiNs, as the peak at ~103.8 eV assigned to $SiO_2$ becomes more prominent after heat treatment, especially in air. Suboxides are also present at lower binding energies between the bulk oxide and the $Si_0$ doublet, which may contain contribution from the lithium silicate structures present in LipSiNs. Similarly, the binding energy of the Li 1s peak is reduced from the initial ~57.2 eV of LiCl to ~55.8–56.2 eV, which can be attributed to lithium oxides forming following heat treatment.

Further XPS analysis before and after surface etching by argon clusters show increasing silicon content towards the core and approximately constant lithium content, resulting in higher surface Li/Si mass ratio compared to the core (Fig. 4a, Supplementary Fig. 9). XPS analysis also shows a trend for reduction in the binding energy of the Li 1s peak (Supplementary Fig. 10) and reduced amounts of oxygen towards the core (Supplementary Fig. 9). These findings suggest an decrease of oxidised lithium species towards LipSiNs core and confirm that the oxidation and lithiation processes occur from the surface. Across all samples, the XPS and ICPMS measurements agree on the relative lithium and silicon content of the nanowires, (Figs. 2e and 4a) confirming the broad range of lithium content achievable with porous silicon lithiation. Chlorine is absent from the XPS survey scans, suggesting that the contribution to lithium content from residual LiCl within the mesoporous structure is minimal (Supplementary Fig. 11).

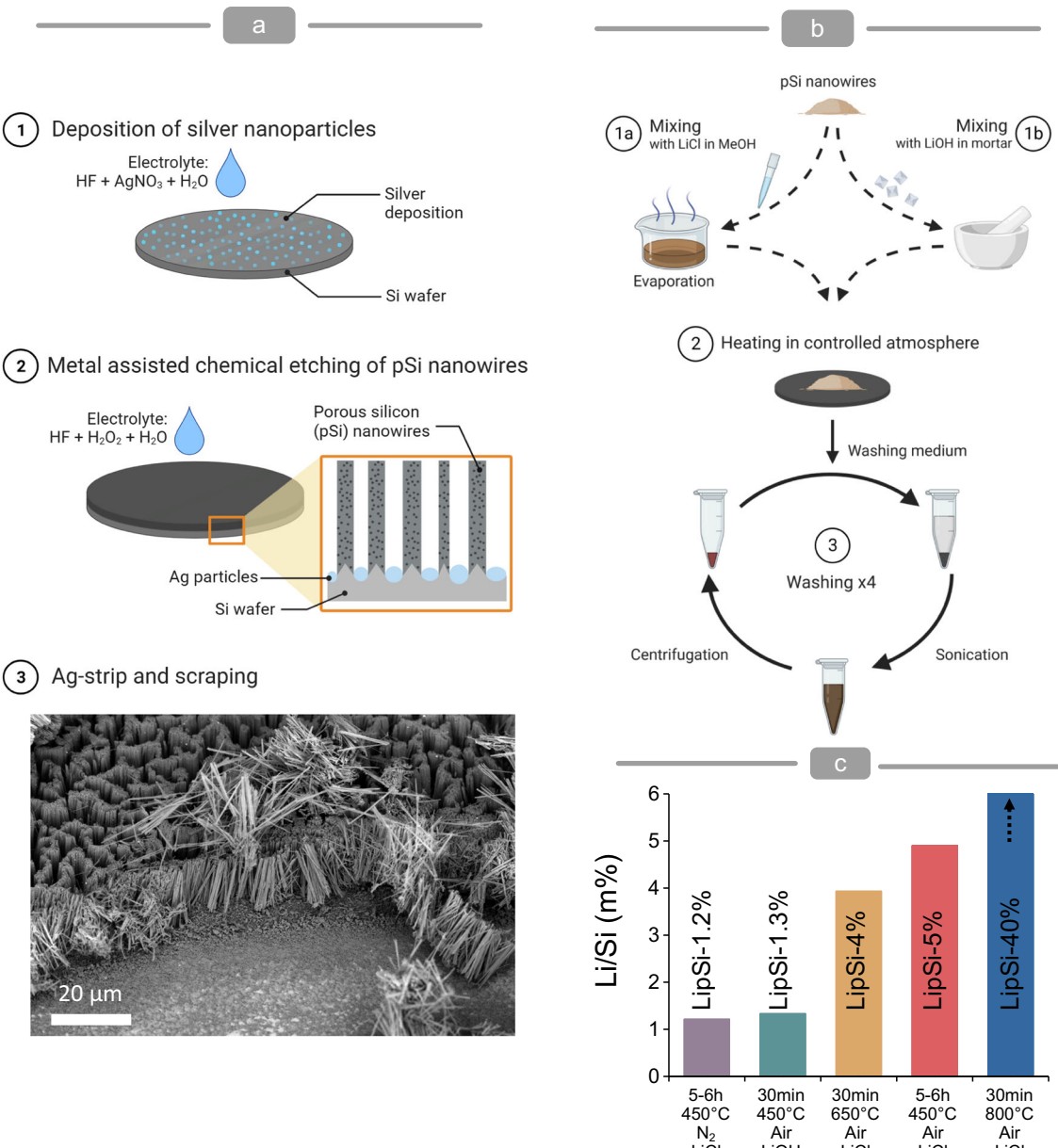

**Fig. 1 | Lithiation of porous silicon nanowires. a** Schematic of the metal assisted electrochemical etching (MACE) process used to generate porous silicon nanowires. (1) A dense network of silver dendrites is deposited by electroless deposition over the surface of a silicon wafer. (2) MACE generates vertically-aligned porous silicon nanowires by etching the silicon substrate in an aqueous solution of hydrofluoric acid and hydrogen peroxide. The resulting porosity depends on the composition of the solution and the resistivity of the silicon wafer. (3) The silver nanoparticles are dissolved and the nanowires collected by mechanical scraping of the silicon wafer. The 45° tilted scanning electron micrograph shows pSiNs being detached from the wafer. **b** Schematic of the lithiation process for porous silicon nanowires. (1) The nanowires are mixed with a lithium precursor either (1a) in a mortar or (1b) in a solution which is then evaporated. (2) The mixed precursors are heated to the desired temperature in a controlled atmosphere which includes the presence of oxygen. (3) Once cooled, the nanowires are washed four times to remove unreacted lithium precursor. The resulting lithiated porous silicon nanowires are ready for use. **c** Quantification of Li/Si ratio of LipSiNs by ICPMS of fully dissolved nanowires as a function lithium precursor, lithiation temperature, time and atmosphere. **a**, **b** Created with biorender.com.

The reduced Li binding energy in LipSi compared to that of the physical mixture of pSi and LiCl further support that the lithium on the surface of LipSiNs preferentially coordinates with the silicon in the nanowire forming lithium silicate structures with varying degrees of oxidation (Supplementary Fig. 8).

The combined EELS, CF3-MALS-ICPMS and XPS data indicate that across all size ranges individual LipSi nanowires are composed of both lithium and silicon. The crystallinity of the nanowires depends on the details and factors of their lithiation process, with LipSi-5% and LipSi 1.2% displaying an amorphous shell with increased lithium content around a crystalline core containing both lithium and silicon, while LipSi-4% are fully crystalline and more homogenous in lithium content throughout.

Raman microspectroscopy of LipSiNs shows a significant blue shift of the Si peak dependent on lithiation conditions, compared to pSiNs undergoing analogous thermal treatments (Fig. 4b, Supplementary Fig. 12). The peak for LipSi-1.3% shifts to 519 cm$^{-1}$ compared to the 515 cm$^{-1}$ for oxidation in air at 450 °C for 30 min, the peak for LipSi-5% shifts to 515 cm$^{-1}$ compared to 512 cm$^{-1}$, and the peak for LipSi-1.2% shifts to 516 cm$^{-1}$ compared to 514 cm$^{-1}$. Non-lithiated, non-thermally treated porous silicon instead shows an expected red shift compared to crystalline silicon (c-Si)[39]. The Raman analysis suggests an

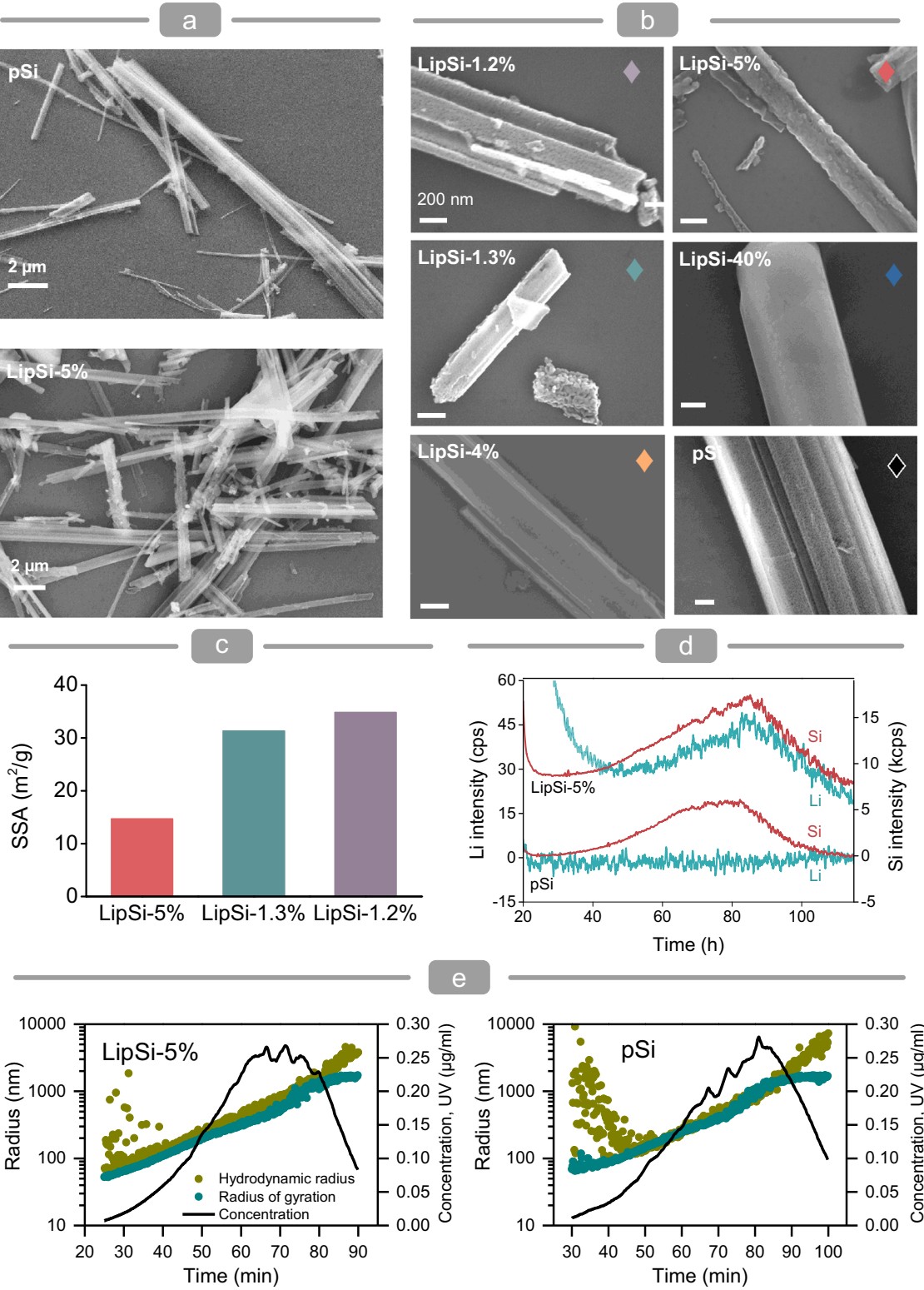

**Fig. 2 | Physicochemical characterisation of lithiated porous silicon nanowires.**
**a** Overview scanning electron micrographs show comparable size distribution of the prepared LipSiNs and precursor pSiNs. Analysis was performed on at least three independent samples for each group. **b** Scanning electron micrographs of LipSiNs and pSiNs showing the morphology and mesoporous structure of the nanowires. Analysis was performed on at least three independent samples for each group.

**c** Specific surface area of LipSiNs from Brunauer–Emmett–Teller (BET) analysis.
**d** Lithium and silicon content of LipSiNs (top) and pSiNs measured in-flow by ICPMS following CF3 separation. Lithium trace in teal blue, silicon trace in red. **e** Radius of gyration, hydrodynamic radius and concentration of LipSiNs and pSiNs measured respectively with in-flow MALS, DLS and UV detectors following CF3 separation.

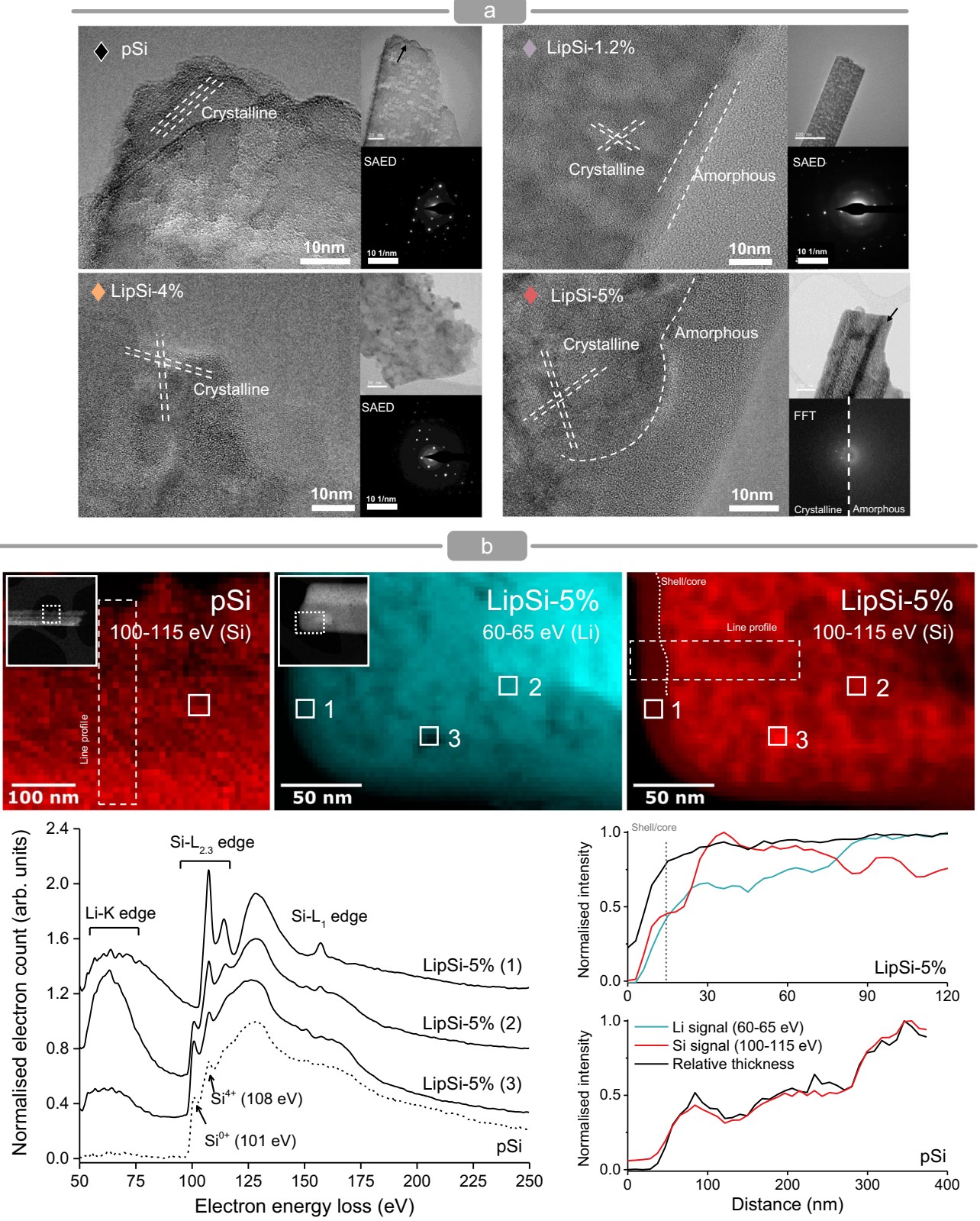

**Fig. 3 | Influence of lithiation on nanowire ultrastructure. a** High resolution transmission electron microscopy of pSiNs and LipSi-4% reveals the crystalline structure of the nanowires extending along the <100> direction, verified with selected area electron diffraction (SAED). LipSi-5% and LipSi-1.2% display a core-shell structure, the FFT pattern of LipSi-5% verifies the single-crystal structure of the core and amorphous phase of the shell. Analysis was performed on five independent nanowires per group. **b** Electron energy-loss spectroscopy spectrum images of Li and Si distribution within pSiNs and LipSi-5% reflecting relative intensity of the Li-K and Si-$L_{2,3}$ edge, respectively. Representative EELS spectra are extracted from the boxed regions. Line profile of Li and Si content alongside relative thickness as quantified from EELS spectra for pSiNs and LipSiNs. The shell to core transition is indicated with a dotted line on the LipSiN graph. Analysis was performed on two independent nanowires per group.

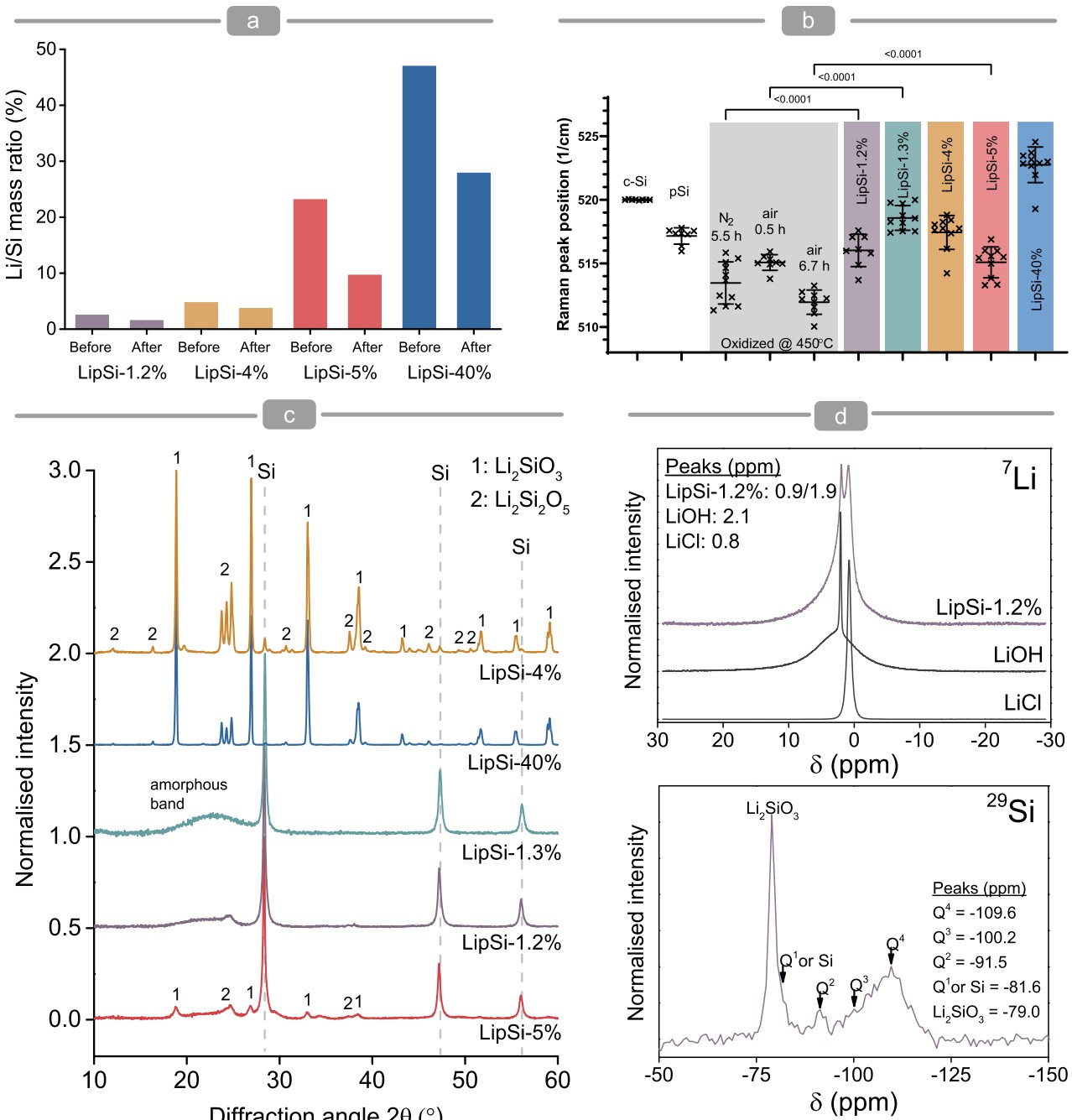

**Fig. 4 | Influence of lithiation on chemical composition of nanowires.**
**a** Quantification of Li/Si content from x-ray photoelectron spectroscopy analysis of the elemental composition of pSiNs and LiSiNs before and after surface sputtering by argon cluster beam. **b** Blue shift in Si Raman peak position for pSiNs and LiSiNs, upon lithiation with respect to comparable thermal treatments for pSiNs. LipSi-1.3% compared to air 0.5 h, LipSi-1.2% compared to N$_2$ 5-6 h and LipSi-5% compared to air 5-6 h. Data are presented as mean values ± S.D. Statistical significance was tested with one-way ANOVA followed by Bonferroni's multiple comparison test. Data was collected from crystalline silicon (c-Si) $n = 7$, pSi $n = 8$, Ox 0.5 h $n = 8$, Ox 6.7 h $n = 9$, OxN2 5.5 h $n = 11$, LipSi 1.2% $n = 8$, LipSi 1.3% $n = 10$, LipSi 4% $n = 10$, LipSi 5% $n = 10$, LipSi 40% $n = 10$ independent measurements. **c** X-ray diffractograms for pSiNs and LiSiNs showing peaks associated with crystalline Si, lithium silicates and amorphous phases within LipSiNs. **d** $^7$Li and $^{29}$Si NMR-MAS spectra of LipSi-1.2% alongside $^7$Li spectra from LiOH and LiCl.

underlying strain in the crystal structure due to lithium coordination. X-ray diffraction highlights the presence of crystalline lithium silicates, Li$_2$SiO$_3$ and Li$_2$Si$_2$O$_5$, which likely contribute to the passivation of LipSiNs via oxidation, and can impact the solubility of LipSiN (Fig. 4c). Several LipSiNs treated at 450 °C also display a broad band between 17° and 27° (LipSi-1.2% and LipSi-1.3%), which is absent in pSiNs treated in comparable conditions, and a broadening of the lithium silicate peaks (LipSi-5%) attributed to the presence of amorphous lithium structures (Supplementary Fig. 13). This amorphous phase likely includes a

contribution from the shell observed by HRTEM and could be a consequence of incomplete annealing due to the lower processing temperatures for these materials (Fig. 3a). NMR-MAS of $^{29}$Si confirms the presence of silicon oxides (Q$^1$-Q$^4$ Fig. 4d) alongside lithium metasilicate (Li$_2$SiO$_3$), while $^7$Li NMR-MAS confirms the presence of multiple species of dielectric Li within LipSiNs.

Silicon L-edge XANES shows a double peak at 102–104 eV associated with crystalline elemental silicon present in the Si wafer reference and in all lithiated substrates. This peak is absent from the

amorphous $Li_xSi_y$ and silicate glass references. Features above 104.2 eV are attributable to surface oxides, consistent with a spectral fine structure similar to $SiO_2$[40] with a shift in the white-line of the lithiated substrates attributable to reduced bond lengths. Lithium L-edge XANES suggests lithiation yields amorphous and crystalline arrangements of Li, Si, and O dependent on degree of lithiation. Experimental spectra are a fit of multiple phases, further supporting the presence of lithium silicate structures (Supplementary Fig. 14).

### Controlled ion release

When exposed to phosphate buffer, LipSiNs do not exhibit visible reactions and dissolve over time analogously to pSiNs, releasing Li and silicate ions including orthosilicic acid[25] (Fig. 5). The modality of lithiation determines the dissolution kinetics of LipSiNs. Overall, lithiation reduces the solubility of LipSiNs with respect to the precursor pSiNs. Lithiated nanowires release 75% of their silicon content (Fig. 5a,b T-75%) within 2 to 4 days depending on the lithiation process, while porous silicon nanowires have a T-75% of 3 hours. The lithiation process controls the degree of burst release for LipSiNs (Fig. 5c). While LipSi-4% shows minimal burst release, LipSi-1.2% releases 45% lithium immediately, and LipSi-1.3% and LipSi-5% reach 74% burst release. Independently of the degree of initial burst release, the subsequent release can last between a few hours for LipSi-5% for up to 4 days for LipSi-1.2%. Comparing the ion release kinetics for LipSi-5% and LipSi-1.2% against LipSi-4%, highlights that the thickness of the amorphous lithium-rich shell correlates with an increase in burst lithium release. The chemical characterisation indicates the concurrent presence of several lithium species within LipSiNs. These species include crystalline lithium silicates which contribute to the stabilisation of the surface of the nanowires and are expected to slow the release kinetics[41,42]. Alongside the more stable silicate species, LipSiNs contain amorphous lithium which contributes soluble material for rapid release. The processing temperature influences LipSiNs crystallinity and in turn regulates the release profile. Indeed, in the high-temperature LipSi-4% (650 °C) the crystalline phase with low-solubility extends all the way to the surface preventing the initial rapid lithium release and slowing silicate ions release compared to pSi. The Li-rich amorphous shell of the low-temperature LipSi-5% (450 °C) and LipSi-1.2% (450 °C) instead contain species that dissolve rapidly contributing to a burst Li release; once the crystalline core is exposed a sustained release occurs analogous to LipSi-4%. The XRD data and the low processing temperature suggest that LipSi-1.3% (450 °C) also possess an amorphous shell and a crystalline core, contributing to its observed dual-release analogous to LipSi-5% and LipSi-1.2%. When LipSi-1.2% and LipSi-5% are exposed to simulated salivary fluid to mimic the oral environment, LipSiNs retain

their tuneable dissolution, displaying a burst early Li release from the amorphous layer, followed by a sustained release from the crystalline structure (Supplementary Fig. 15). Silicon release rate is slowed with respect to phosphate buffer, due to the reduced hydrolytic process in the lower pH environment[26]. Overall, the ion release data indicates that our process can regulate the introduction of lithium within porous silicon to achieve a tuneable, controlled release of lithium and silicon in simulated biological fluids. Since the temporal modulation of the Wnt/β-catenin stimulus is important to determine its regenerative effect, the tunability of lithium release provided by LipSiNs can contribute to optimise the activation profile of Wnt/β-catenin.

### In-vitro bioactivity

Investigating the effect of LipSiNs-conditioned medium (LipSiN-CM) on human periodontal ligament stem cells (hPDLSCs) enables rapid screening in a simple, patient-relevant model, to identify the lithiation conditions with the most promising osteogenic and Wnt/β-catenin activation capacity for periodontal regeneration. Incubating LipSiNs with cell culture medium to prepare LipSiN-CM does not affect the pH of the medium (Supplementary Fig. 16) in solubility limit conditions (Supplementary Fig. 17). Lithiation improves the cytocompatibility of the nanowires, since hPDLSCs retain viability after 24 h in LipSiN-CM up to 1.6 mg/ml, while pSiN-CM shows cytotoxic effects starting from around 0.8 mg/ml (Fig. 6a, b). The ability of LipSiN-CM to stimulate *AXIN2* gene expression and activate the Wnt/β-catenin pathway depends on LipSi nanowires formulation. LipSi-1.2% shows dose-dependent *AXIN2* activation after 24 h, which matches the stimulus provided by 5 mM LiCl at 1.6 mg/ml (Fig. 6c). Conversely, LipSi-5% does not exhibit *AXIN2* activation, similarly to pSiNs. As expected from the osteogenic activity of pSi, both pSiN-CM and LipSiN-CM upregulate key osteogenic markers (OCN, BMP, RUNX2) in hPDLSCs over 14 days, at levels comparable to osteogenic medium (Fig. 6d). Interestingly for periodontal regeneration, LipSiN-CM and pSiN-CM also upregulate the cementogenic marker (CAP) over the same timeframe.

LipSiN in direct contact with cells also stimulate Wnt/βcat and osteogenesis (Supplementary Fig. 18). Incubation with LipSiNs induces toxicity at concentrations above 50 μg/ml, while upregulating *AXIN2* expression and stimulating osteogenesis at concentrations above 10 μg/ml. In this setup, cells are directly exposed to LipSiN dissolving for up to 14 days, indicating that the ion release profile has a stimulatory effect towards the Wnt/βcat pathway and osteogenesis over the desired timeframe. These data indicate that lithiation improves the biocompatibility of pSiNs, while the details of the concentration and release profile of lithium play an important role in determining GSK3 inhibitory activity for LipSiNs in cell models relevant for human

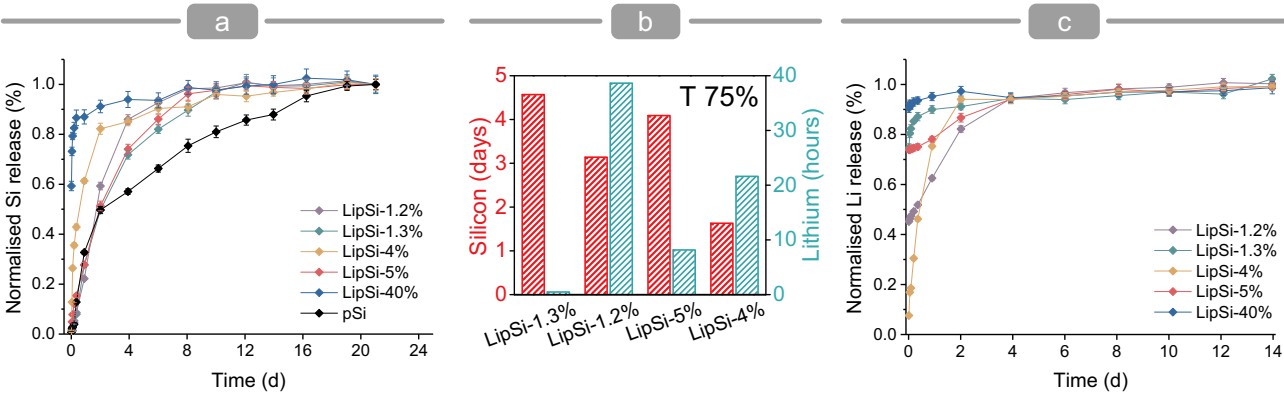

**Fig. 5 | Lithiation modulates ion release. a** Si release profile quantified by longitudinal inductively copuple plasma mass spectrometry (ICP-MS) from nanowires dissolving in phosphate buffer solution; **b** estimated time to achieve 75% of total ion release as calculated from profiles (**a, c**); **c** Li release profiles for LipSiNs quantified by longitudinal ICPMS from nanowires dissolving in phosphate buffer solution. **a, c** Data are presented as mean values ± S.D. Data was collected from *n* = 3 independent measurements.

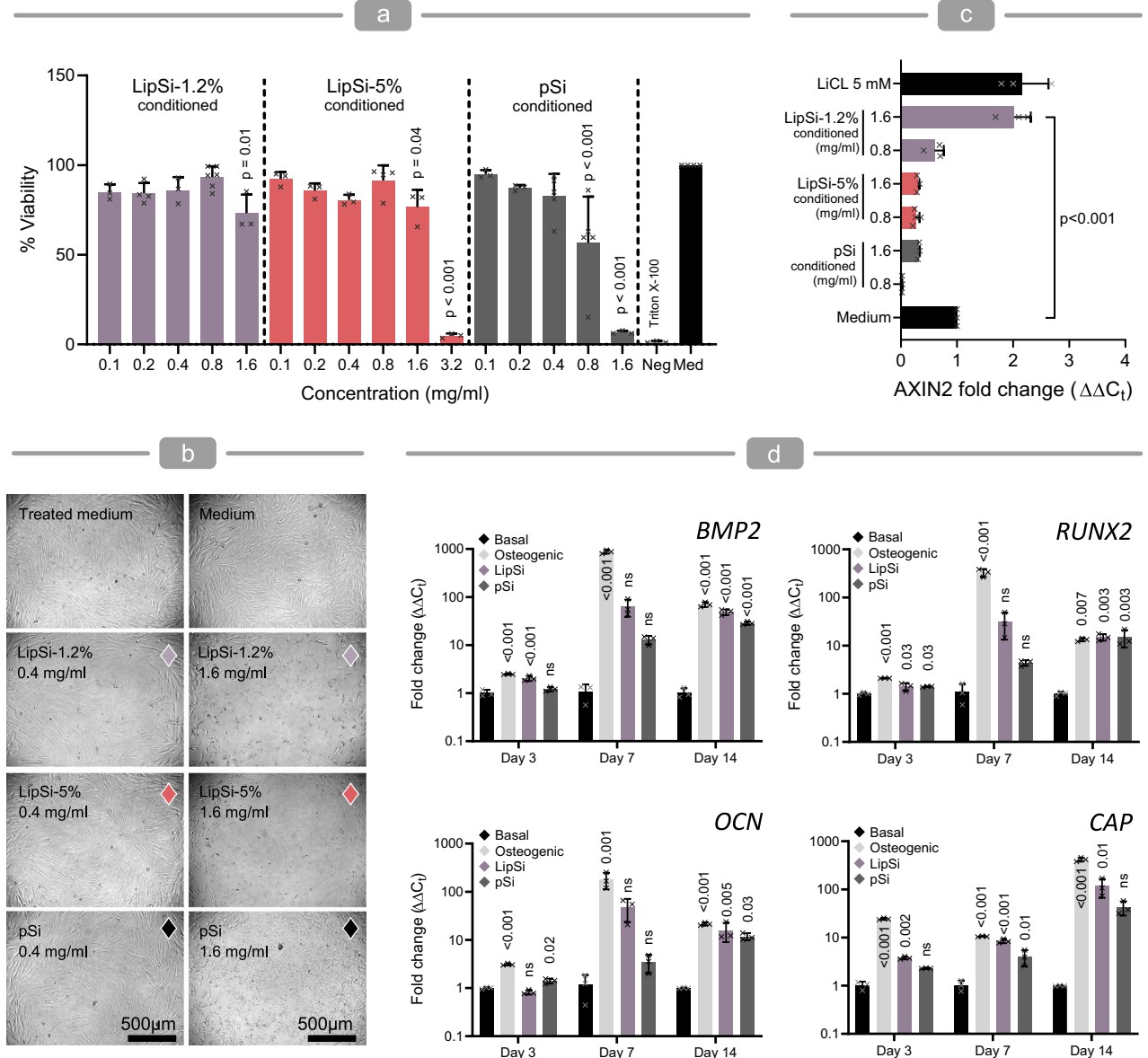

**Fig. 6 | Lithiated porous silicon nanowires are biocompatible and bioactive.**
**a** Metabolic activity for hPDLCs cultured using pSiNs and LipSiNs conditioned medium as determined by ATP activity assay. Med group represents control medium which was thermally treated without nanowires, Neg group represents cells treated with Triton X-100. Data are presented as mean values ± S.D. Statistical significance was tested with one-way ANOVA followed by Dunnett's multiple comparisons post-hoc test. $n = 3$ biologically independent samples were examined in all cases except: $n = 4$ in LipSi-1.2% (0.2 mg/ml), LipSi-5% (0.8 mg/ml) and Med; $n = 5$ in pSi (0.4 mg/ml and 0.8 mg/ml); $n = 6$ in LipSi-1.2% (0.8 mg/ml). $p$-values are reported with respect to control medium. **b** Bright field images of hPDLCs cultured in conditioned media for 24 h. **c** Relative expression of *AXIN2* as quantified by qPCR for hPDLSCs cultured in media conditioned with different types of LipSiNs. Medium

with the addition of 5 mM LiCl serves as positive control for *AXIN2* expression while non-conditioned medium is used as reference. Data are presented as mean values ± S.D. Statistical significance was tested with one-way ANOVA followed by Tukey's multiple comparison post-hoc test. $N = 3$ independent biological samples. **d** qPCR analysis of relative expression of osteogenic (*BMP2, RUNX2, OCN*) and cementogenic (*CAP*) genes for hPDLSCs cultured in media conditioned with pSiN or LipSiN-1.2%. Basal medium serves as reference for non-stimulated cells, while osteogenic medium serves as positive control. Data are presented as mean values ± S.D. Statistical significance was tested with one-way ANOVA followed by followed by Tukey's multiple comparison post-hoc test. $N = 3$ independent biological samples. $p$-values are reported with respect to basal medium gene expression for the same timepoint.

periodontal regeneration. Overall, LipSi-1.2%, a slow lithium-releasing formulation (Fig. 5), displays the combination of Wnt/β-catenin and osteogenic stimuli desired for in vivo regeneration of periodontal defects.

## Periodontal regeneration

Preliminary testing of LipSiNs in mouse models enabled determining the efficacy of their controlled release for Wnt/β-catenin stimulation in vivo (Fig. 7). LipSiNs injected from solution within the periodontal

pocket of mice remain in situ for up to 24 h (Fig. 7a). The injection and retention of the LipSiNs does not affect the structure or the morphology of the tissue. When injected in the periodontal ligament of an Axin2^LacZ/LacZ murine reporter, LipSiNs show periodontal *AXIN2* activation within 24 h, demonstrating in vivo Wnt/β-catenin stimulation (Fig. 7b).

A rat model of periodontal fenestration defect involving alveolar bone, periodontal ligament, and tooth was used to determine the efficacy of LipSiN treatment for periodontal regeneration (Fig. 8a). The

animals were treated using nanowires incorporated within a pluronic F-127 hydrogel carrier, necessary to fill the defect void (supplementary information §1.20.1). Based on the efficacy and biocompatibility data gathered from periodontal cells in vitro, LipSi-1.2% at 1.6 mg/ml concentration were selected as the treatment group. The control groups included no treatment (Fig. 8b), as well as the use of the hydrogel carrier in combination with: pSiNs at 1.6 mg/ml concentration to evaluate the effect of lithium over silicon, lithium chloride at 25 mM concentration to evaluate the effect of combining lithium with silicon and the effect of sustained lithium release, and a commercial guided tissue regeneration membrane (GTR, BioGide®) used for periodontal regeneration in the clinic as a gold standard. At two weeks post-treatment μCT shows that only LipSiN treatment improves bone mineral density (BMD), by 119% compared to untreated control, while both LipSiN and LiCl improve bone volume over total volume (BV/TV) by 78% and 64%, respectively, over control (Fig. 8b–d). Furthermore, at two weeks, LipSiNs improve BMD over GTR by 32%. The six-week analysis analogously shows that LipSiNs increase BMD by 34% over control and 33% over GTR, while BV/TV trends 21% higher than control and 12–18% higher than other groups (Fig. 8b–d). The μCT data indicate that LipSiN treatment accelerates bone regeneration in periodontal defects.

Histological analysis of H&E tissue sections collected at two and six weeks following treatment confirms the bone regeneration observed by μCT (Fig. 9a, b). Furthermore, histology highlights LipSiN's effective cementum regeneration, shown by the appearance of new cementum in the vicinity of the regeneration site (Fig. 9a, b, black arrowheads at new cementum interface in LipSi-1.2% group). Masson staining indicates that LipSiN induce significant formation of cell-laden, collagen-rich, non-mineralised tissue in the interstitium between the newly formed bone and new cementum, where the periodontal ligament compartmentalizes. The newly formed fibres in the LipSiN group have regular orientation and are embedded within the new cementum and new bone, unlike the untreated group. The presence of both CD31+ vascular endothelial cells and Osterix+ osteoblasts within newly-formed bone only for LipSiNs stimulation, indicates their ability to establish vascularised bone (Fig. 10a, Supplementary Fig. 19). Perivascular α-SMA-expressing progenitors are hypothesised to differentiate into osteoblasts, cementoblasts and fibroblasts during periodontal regeneration[43]. The newly formed bone in the LipSiN group includes α-SMA expressing cells, suggesting a possible link to α-SMA+ mediated regeneration. Elemental mapping of histological samples by laser ablation inductively coupled plasma mass spectrometry (LA-ICP-MS) does not evidence additional silicon or lithium within the newly-formed tissue, indicating that released ions stimulate regeneration but are not incorporated within the resulting tissues (Fig. 10b). The histological presentation of the liver and kidney of LipSi-treated animals was comparable to the control group, confirming the lack of toxicity to organs beyond the oral cavity (Supplementary Fig. 20).

We also compared the regenerative capability of LipSiN to that of bioglass where the sodium content was fully substituted with lithium (Li-BG), as a model of lithium and silicate ion release with an established stimulatory capacity[44–46] (Supplementary Figs. 21–23). MicroCT analysis showed that both Li-BG and LipSiN stimulated the regeneration of the bone defect, with LipSiN outperforming Li-BG for both bone volume and bone mineral density at 2 weeks and 6 weeks (Supplementary Fig. 21). Histological analysis revealed that LipSi and Li-BG stimulated the regeneration of new cementum and soft tissue (Supplementary Fig. 22) and immunohistochemistry showed the formation of comparably vascularised tissue for LipSi and Li-BG (Supplementary Fig. 23). Overall LipSiN demonstrated a comparable effectiveness to Li-BG with a capability for bone regeneration.

These data indicate that LipSiNs stimulate Wnt/β-catenin in vivo and display a regenerative activity towards bone, cementum and periodontal ligament in our model of periodontal defect. This regenerative potential is significantly higher than that of lithium chloride, porous silicon nanowires, lithium-substituted bioglass, and a commercial GTR membrane used for the treatment of periodontitis.

## Discussion

Overall, lithiated porous silicon has broadly tuneable physicochemical properties, enabling tight control over ion release kinetics, and in turn determining their bioactivity, which can be tailored to provide effective regenerative stimuli. Porous silicon nanowires produced by metal assisted chemical etching are a compelling material for biomedical applications. Conventional silicon nanowire manufacturing processes, such as vapour-liquid-solid growth, require high-temperature processes, highly controlled environments and high purity starting materials, leading to limited scalability and high costs. Metal assisted chemical etching instead is a room-temperature, solution-based process that can use low grade and bio-sourced silicon to yield large quantities of porous silicon at competitive costs[37,47,48]. Surface passivation through lithium and silicon species provides a surprising combination of non-reactivity, improved biocompatibility and retained degradability. This tailorability and bioactivity of lithiated porous silicon promise to extend the range of application for porous silicon-based materials. This work demonstrated the suitability of lithiated porous silicon for the regeneration for complex structures comprising mineralised and soft tissue, and tissue interfaces. Beyond tissue regeneration, the controlled activation of Wnt/β-catenin provided by the sustained lithium release can find applications in chronic wound healing, and the treatment of neuropsychiatric disorders and neurodegenerative diseases. Moving forward, in order to broaden the use of lithiated porous silicon across biomedical applications that can benefit from sustained lithium release, it is crucial to gain a systematic understanding of the effect that lithiation parameters have on its physicochemical properties, and to further investigate its biocompatibility for a broader range of cells and tissues. Specifically for periodontal regeneration, tailoring LipSiNs formulation towards the broad range of clinical presentations of periodontal defects is essential to bring the technology to the clinic.

## Methods
### Metal assisted chemical etching (MACE)
Porous silicon nanowires were etched via metal assisted chemical etching (MACE) as described in our previous publications[1]. A schematic representation of the MACE process can be found in Fig. 1a. Briefly, prime grade 100 mm silicon wafers, with boron doping, 0.01-0.02 Ωcm resistivity, and (100) orientation, were purchased from University Wafers Inc (USA). Wafers were first cleaned from native surface oxide by dipping them into 1:4 mixture of hydrofluoric acid 50% (Honeywell 40213H) and deionised (DI) water (10% HF). Cleaned wafers were rinsed with DI water and isopropanol and dried in $N_2$ flow. After cleaning, wafers were immersed into solution of 20 mM silver nitrate ($AgNO_3$, Sigma-Aldrich 99.9999%) in 10 % HF for 2 min under constant gentle stirring. Wafers were rinsed and dried as before. Etching of porous silicon nanowires was done by immersing silver deposited wafers for 20 min in an etching solution containing 0.6 wt% (or 1.2 wt%) of $H_2O_2$ (Honeywell, 95299) in 10% HF. Etched wafers were rinsed and dried as before. As a last step, deposited silver particles were removed by 10 min treatment with standard gold etchant (Aldrich, 651818). Etched wafers were rinsed and dried as before. Prior the use, nanowires were collected by scraping with razor blade.

### Lithiation
Schematic representation of the lithiation process can be found in Fig. 1b. Porous silicon nanowire powder was mixed with lithium containing agents: LiCl (Fluorochem, 091451), LiOH*$H_2O$ (Fluorochem, 009861) or $Li_2CO_3$ (Acros Organics, 197785000). Two different mixing processes were used, grinding for LiOH and $Li_2CO_3$ or liquid evaporation for LiCl. In the grinding method, a pre-weighted pSi nanowire

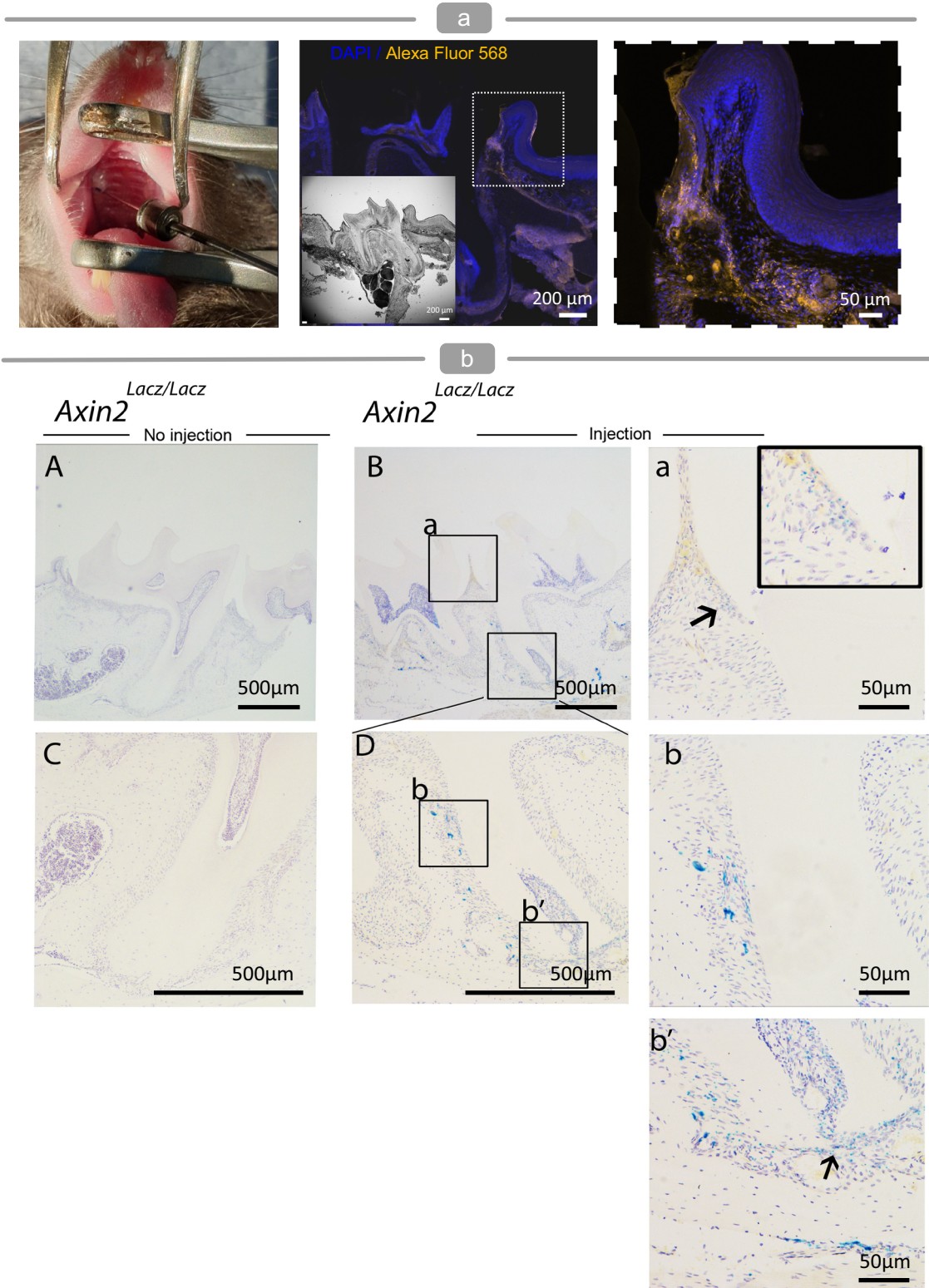

**Fig. 7 | LipSiNs stimulate Wnt/β-catenin signalling in vivo.**
**a** Immunofluorescence analysis of histological slides of the mandible of a mouse 24 h post-injection of LipSi−1.2% in the periodontal space. LipSi−1.2% labelled with Alexa Fluor 568 in orange, DAPI in blue. Analysis was performed on 4 animals.

**b** Immunohistochemical analysis of Axin2 expression in histological slides of the mandible of an *Axin2*[LacZ/LacZ] mouse 24 h following post-treatment with LipSi−1.2% compared to untreated control. The punctate blue signal (shown by black arrows) indicates Axin2-expressing cells. Analysis was performed on 4 animals.

powder (brown) and solid lithium source (white) were combined in a mortar (Supplementary Fig. 2). The particle size of the lithium source and agglomerate size of pSi nanowires was reduced by gently grinding with a pestle. Mixing/grinding was done until the powder appeared

homogenous in structure and colour. The powder was then poured on top of a silicon wafer, covered with a steel vial, and moved on a hot-plate preheated to 450 °C. In the solvent evaporation method, LiCl was dissolved into methanol at 40 mg/ml concentration. pSi nanowire

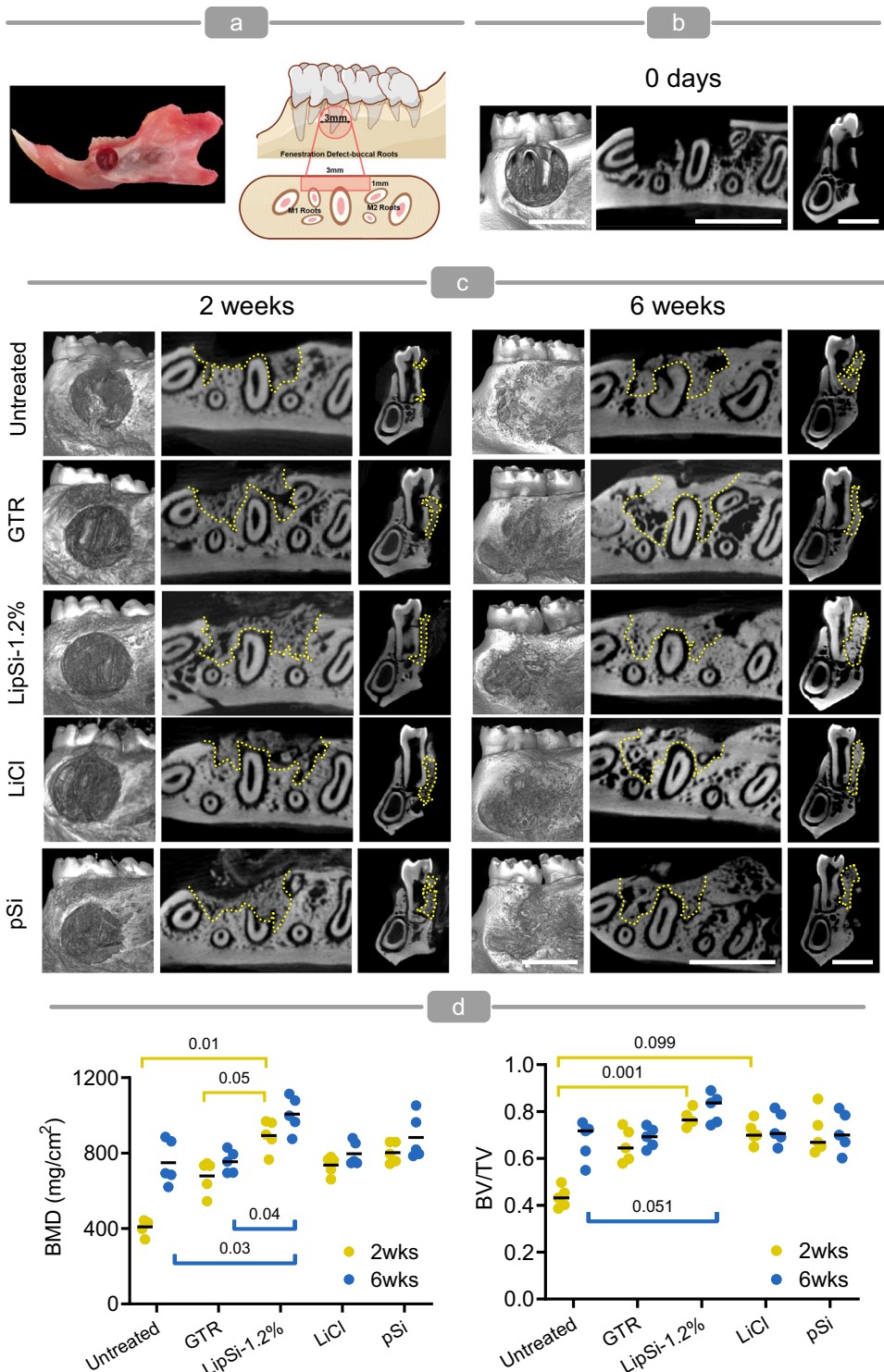

**Fig. 8 | LipSiNs promote bone regeneration in periodontal defects. a** The model of periodontal fenestration defect (standardised with 3 mm in length, 3 mm in height and <1 mm in deep) used in this study. **b** μCT scans of rat mandibles at day 0. Scale bar 3 mm. Analysis was performed on 5 animals. **c** μCT scans of rat mandibles showing regeneration of periodontal defects 2-weeks and 6-weeks post-operative with lithium chloride, BioGide® GTR membrane, pSi, and LipSi−1.2%; serves as baseline comparison. The dotted yellow line outlines the newly formed bone. Scale bar 3 mm. Analysis was performed on 5 animals per group. **d** μCT analysis for the quantification of BV/TV and BMD. Data are presented as mean values ± S.D. Statistical significance was tested by Kruskal–Wallis non-parametric multivariate analysis followed by Dunn's multiple comparison post-hoc test. *N* = 5 independent biological samples.

powder and LiCl solution were combined in mortar, mixed briefly with the pestle, and moved to a hotplate preheated to 100 °C. After the methanol had evaporated, the temperature was increased to 150 °C for 10 min to enable evaporation of any remaining solvent or moisture. The solid mixture of pSi and LiCl was moved onto a Si wafer preheated to 150 °C and made into a fine powder. The Si wafer was moved onto a hotplate preheated to 450 °C and covered immediately with a bell jar. If treatment was performed in ambient atmosphere, the top cap was left open, if the treatment was performed in $N_2$ atmosphere, $N_2$ flow was directed into bell jar through the top valve.

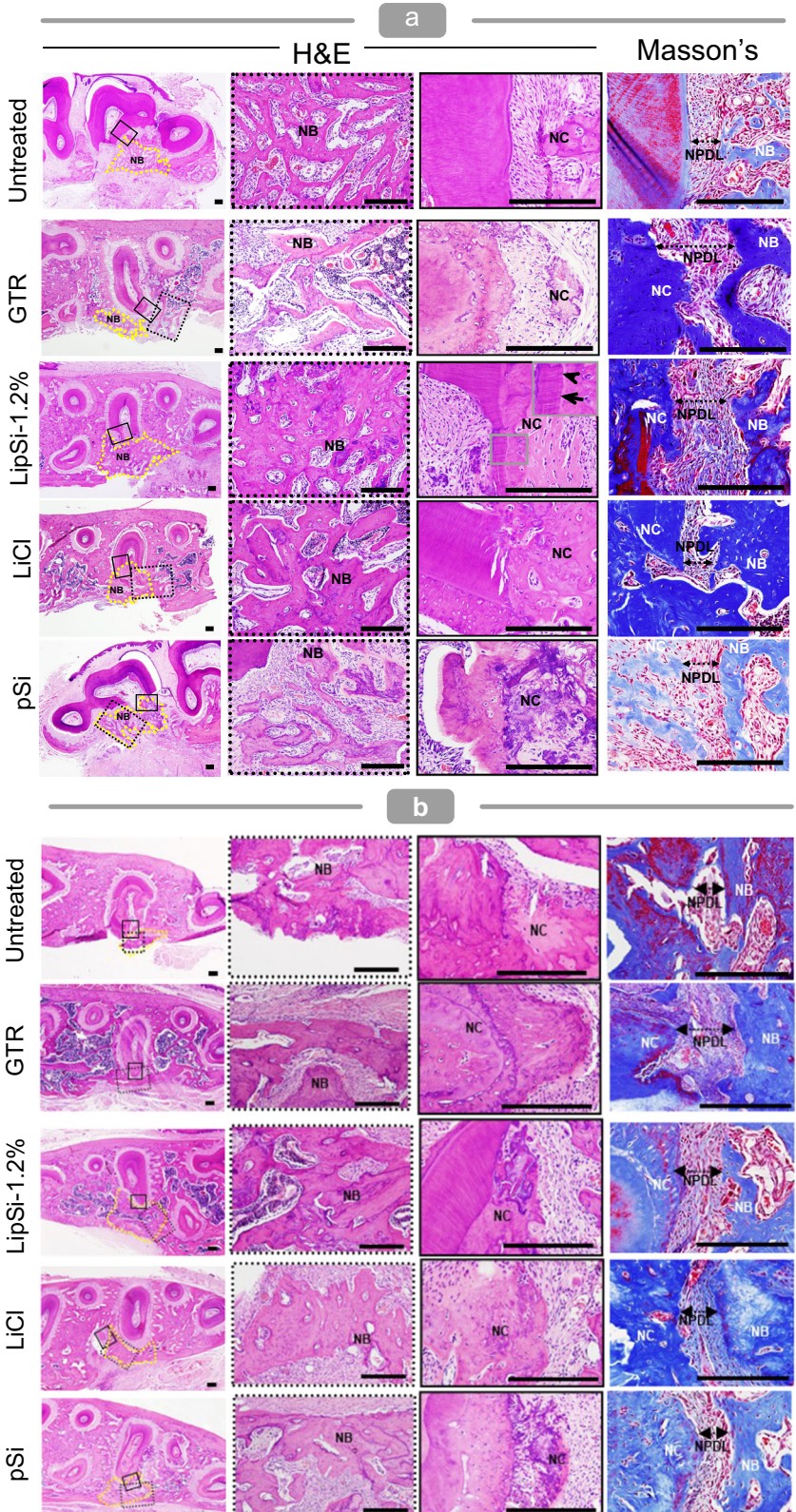

**Fig. 9 | LipSiNs stimulates regeneration of mineralised and soft tissue in the periodontium. a** Histological analysis of alveolar bone, cementum and periodontal ligament regeneration at 2 weeks and (**b**) 6-week post-operative. Three left panels H&E staining, rightmost panel Masson's Trichrome staining. Leftside panel shows an overview of the periodontal defect and its regeneration. Within the leftside panel yellow dotted line indicates newly formed bone; black dotted line box locates the magnification panel used to visualise bone regeneration; black solid line box locates the magnification panel used to visualise cementum regeneration. Rightmost panel indicates periodontal ligament regeneration and its integration with bone and cementum. NB new bone, NC new cementum, NPDL new periodontal ligament. Black arrow heads indicate the interface of the newly formed cementum on the root dentin. Scale bars 200 μm. **a**, **b** Analysis was performed on 5 animals per group.

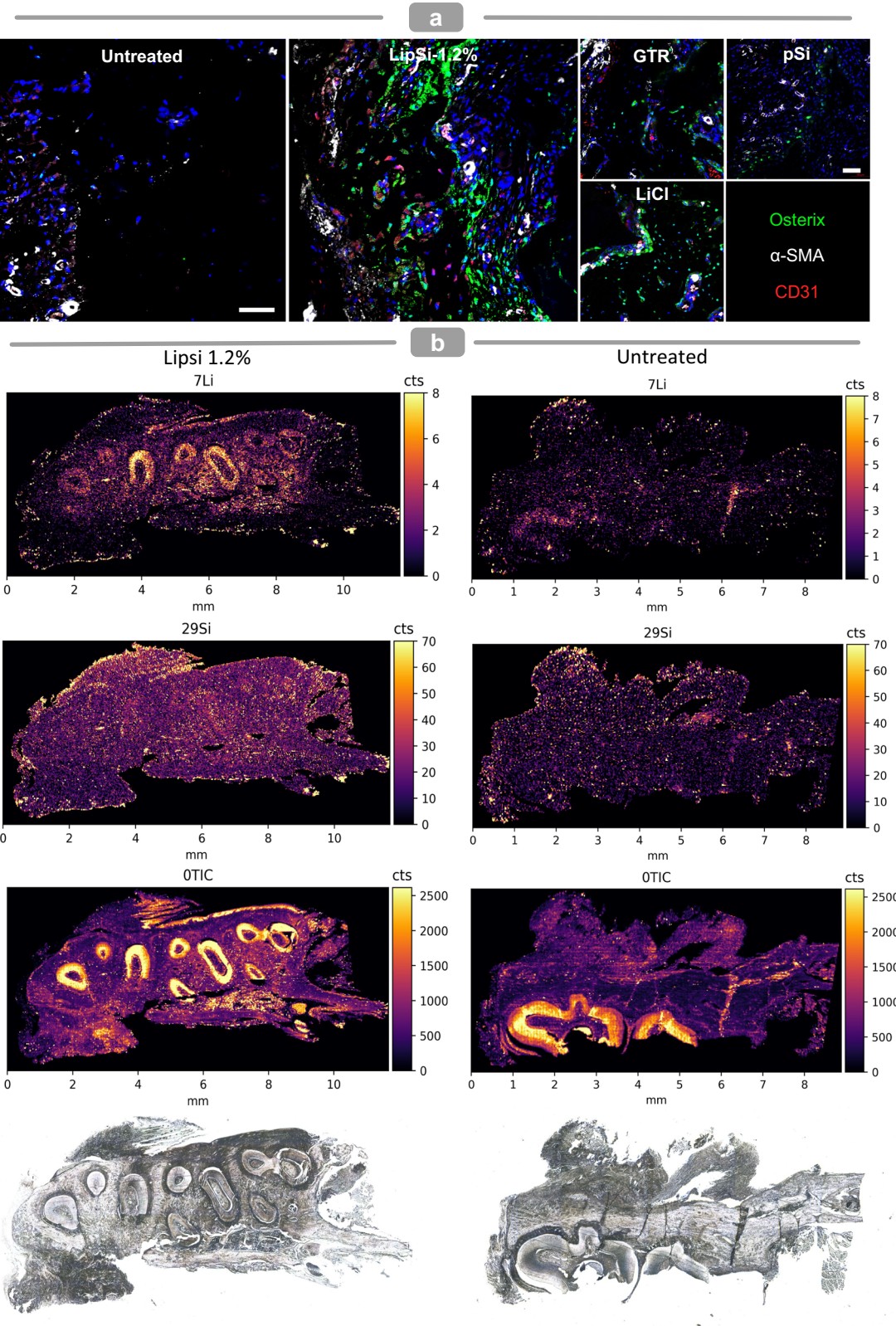

**Fig. 10 | Molecular and chemical characterisation of LipSiN-regenerated periodontium. a** Co-immunofluorescence staining of histological slides with Osterix (green), α-SMA (white) and CD31 (red) in the area of the regenerated periodontium. Scale bars 50 µm. Analysis was performed on 5 animals per group. **b** Elemental mapping of histology slides with LA-ICP-MS, TIC represents the total ion count for all elements imaged including $^7$Li, $^{29}$Si, $^{31}$P, $^{55}$Mn, $^{57}$Fe and $^{88}$Sr. The figure axis indicate distance in mm.

After lithiation, the excess lithium precursor was washed by dispersing the nanowires into dilute 1 mM HCl (in case of the LiOH precursor 1 M HCl was used), mixed with a sonicator bath, centrifuged for 8 min at 18,000 × g, and the supernatant was then discarded. The washing cycle was repeated 4 times with different media (HCl, H$_2$O, isopropanol, isopropanol). After the final wash, nanowires were dispersed into isopropanol with a sonicator bath and stored at RT.

The concentration of the LipSi suspension was determined by weighting the solid content. 0.5 ml of LipSi suspension was pipetted into a pre-weighted 1.5 ml microcentrifuge tubes, nanowires were spun down, isopropanol was discarded and the tubes were dried at 60 °C for at least 30 min. The caps were closed and the tubes were allowed to reach the original temperature and moisture level by keeping them at RT for at least 15 min prior weighing again. At least two microcentrifuge tubes were use per one sample and the weighing was performed at least 3 times.

### Reporting summary

Further information on research design is available in the Nature Portfolio Reporting Summary linked to this article.

## Data availability

All data presented in the manuscript is available from the authors upon request.

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

## Acknowledgements

C.C. acknowledges funding from the European Research Council Starting Grant award 'ENBION' (759577), from the Medical Research Council Confidence in Concept award (MC_PC_18052). We acknowledge funds from the EPSRC (EP/R02863X/1) for access to the Leeds EPSRC Nanoscience and Nanotechnology (LENNF) Facility for electron microscopy, and from the King's College London – Peking University Health Science Center joint initiative for Medical Research. R.Z. acknowledges funding from The National Natural Science Foundation of China(82270945). The X-ray photoelectron (XPS) data collection was performed at the EPSRC National Facility for XPS ("HarwellXPS"), operated by Cardiff University and UCL, under Contract No. PR16195. P.S. acknowledges support from the Czech Science Foundation, project GACR 21-21409S. We acknowledge beamline VLS-PGM at the Canadian Light Source (proposal number 10662) for access to their instruments resulting in data presented within this study. We acknowledge the Materials Research Infrastructure (MARI) at Department of Physics and Astronomy, University of Turku for access and support with the SEM, XPS and XRD facilities.

## Author contributions

Author contributions based on CRediT taxonomy. M.K.: Conceptualisation, methodology, investigation, formal analysis, Writing – original draft, Visualisation, funding acquisition. R.Z.: methodology, formal analysis, Writing – Review & editing, Visualisation, funding acquisition. P.V.: methodology, investigation, formal analysis, Writing-review and editing. A.A.B.: investigation, formal analysis, writing – Review & editing. Ma.S'A.: investigation, formal analysis, writing – Review & editing. D.M.: investigation, formal analysis. M.I.: methodology, investigation, formal analysis, visualisation, writing – Review & editing, funding acquisition. E.M.: investigation, formal analysis. C.W.: investigation. E.M.: methodology, investigation, formal analysis. P.C.: resources, writing – review & editing. A.P.: investigation, formal analysis. V.C.: investigation. X.Z.: investigation. S.A.M.: investigation. A.P.M.: methodology, investigation, formal analysis. E.G.: resources, review and editing. D.S.B.: resources, review and editing. O.A.: methodology, formal analysis, review & editing. X.Z.: methodology, investigation. A.A.: resources, writing – review & editing. M.B.: methodology, resources, formal analysis, writing – review & editing. K.A.-J.: resources, writing – review & editing. J.S.: resources, formal analysis, writing – review & editing. N.H.: resources, formal analysis, writing – review & editing, funding acquisition. P.S.: conceptualisation, methodology, resources, funding acquisition, writing – review & editing. C.C.: conceptualisation, methodology, formal analysis, resources, Writing – original draft, supervision, project administration, funding acquisition.

## Competing interests

M.K., P.S. and C.C. declare a financial competing interest as inventors of patent application EP4192788A1 currently at PCT stage and assigned to King's College London. The patent application covers the method to obtain lithiated porous silicon nanomaterials presented in this manuscript. No other authors have competing interests.

## Additional information

[1]Centre for Craniofacial and Regenerative Biology, King's College London, London SE1 9RT, UK. [2]Department of Oral Pathology, Peking University School and Hospital of Stomatology, Beijing 100081, PR China. [3]School of Chemical and Process Engineering, University of Leeds, Leeds LS2 9JT, UK. [4]Department of Chemistry, University College London, London WC1H 0AJ, UK. [5]HarwellXPS, Research Complex at Harwell, Rutherford Appleton Labs, Didcot OX11 0DE, UK. [6]Department of Physics and Astronomy, University of Turku, Turku 20014, Finland. [7]Postnova Analytics GmbH, Rankinestr. 1, Landsberg am Lech 86899, Germany. [8]Department of Materials, Imperial College London, London SW72AZ, UK. [9]The Francis Crick Institute, London NW11AT, UK. [10]School of Veterinary Medicine, Faculty of Health and Medical Sciences, University of Surrey, Guildford GU2 7XH, UK. [11]Centre for Oral Clinical & Translational Sciences, King's College London, London SE1 9RT, UK. [12]Otto Schott Institute of Materials Research, Friedrich Schiller University Jena, Jena 07743, Germany. [13]Department of Dental Materials & NMPA Key Laboratory for Dental Materials, Peking University School and Hospital of Stomatology, Beijing 100081, PR China. [14]Institute of Pharmaceutical Science, King's College London, London SE1 9NH, UK. [15]Institute of Animal Physiology and Genetics, Czech Academy of Sciences, Brno 602 00, Czech Republic. [16]London Centre for Nanotechnology, King's College London, London WC2R 2LS, UK. [17]These authors contributed equally: Martti Kaasalainen, Ran Zhang ✉e-mail: ciro.chiappini@kcl.ac.uk

