## [Peer Review File · Nature Communications]

Lithiated porous silicon nanowires stimulate periodontal regenerationReviewers' Comments:

Reviewer #1:

Remarks to the Author:

This manuscript describes the preparation of lithium-doped porous silicon nanowires, the release of lithium ions and silicate, the cell compatibility of the material in indirect contact and the osteoinductive behaviour in terms of promoting bone regeneration in periodontal defects.

The manuscript is well written and uses the appropriate methods for materials characterisation. The rat in vivo model also seems appropriate.

My concerns with this manuscript are a) the lack of or at least confusing explanation on whether we are dealing here with lithium silicide or silicate or a mix, and b) the lack of comparison with lithium bioceramics already described in the literature for the same application.

My detailed comments are as follows:

- The introduction is nice and it suggests that Li_xSi_y materials used in lithium batteries are flammable and reactive. The lithium incorporation approach used here probably produces lithium silicate or lithium in ionic form which of course is not flammable. That indicates that this critical difference is not fully appreciated by the authors.
- That leads me to the results part where the authors put a lot of effort into characterising their materials. But they do seem to draw inconsistent conclusions. In some parts they talk about lithiation indicating the formation of lithium silicide (XPS, EELS) and lithium silicate in other parts (Raman, NMR-MAS, XANES). This would need to be teased out by further studies including using high resolution XPS.
- The material should also be characterised before and after heat treatment at 450 deg C.
- The absence of chlorine signal in XPS (from LiCl) could still mean that lithium ions are adsorbed to the oxidised silicon surface.
- On page line 184 the authors talk about Si ions being released. I assume they mean silicate ions.
- The influence of the heat treatment on lithium ion release should also be presented.
- As for the in vivo model and the overall novelty of the work, it would be important to study any significant differences in terms of the performance compared to other lithium doped bioceramics (including silicon based ones like MSN) which have already been studied for the same purpose. The benchmarking against GTR is useful but not sufficient in my opinion.
- Is there a good justification of using porous silicon nanowires which are more difficult and expensive to produce at scale?

Due to the lack of clarity in material characterisation and the lack of a clear and significant advance I would not recommend accepting the manuscript in its current form.

Reviewer #2:

Remarks to the Author:

The paper by Kaasalainen et al. developed lithiated porous silicon nanowires for periodontal regeneration. Authors demonstrated the fabrication process and its characterization using a variety of techniques to support their conclusions. Overall this work is very appealing for NC and deserves consideration.

Considering the application of these nanowires for biomedical purposes, I'd recommend some more additional experiments before publication.

1. Please provide the release studies in Fig. 5a also in simulated oral/salivary fluids. This better represents the biomedical application.

2. The release of both ions is from few days to several hours. Authors should demonstrate that the amounts in such times have a therapeutic effect or enough to lead to a therapeutic outcome.
3. Authors have to demonstrate that there are no adverse effects in case the nanowires end-up elsewhere besides the oral cavity.
4. Authors have to demonstrate by using a control of a mixture of equal amounts of Si and Li found in the nanowires also as an hydrogel formulation to clear prove the nanowire is needed for the intended application.
5. In addition, a control using a commercial available product for this medical application should be included in the study to demonstrate the benefit of this novel approach compared to existent ones.

Reviewer #3:

Remarks to the Author:

In this manuscript, the authors reported a kind of "lithiate porous silicon nanowires (LipSiNs)" strategy to generate a biocompatible and bioresorbable material for treating Periodontal disease. Prelithiation, a strategy developed for battery technology, is proposed via the technology to stimulate bone repair via silicic acid release while providing regenerative stimuli through lithium activation of the Wnt/ β -catenin pathway. Therefore, the unique structure exhibited a good treatment method. However, there are still some problems as follows before published:

1. To date, SiNWs were fabricated by vapor-liquid-solid (VLS) process, which has not been possible to prepare in batches. Please compare with the preparation method of LipSiNs, including cost, preparation process, and yield, and explain the advantages of the work.
2. The size distribution of LipSiNs should be provided, including the length and diameter of nanowires.
3. HRTEM characterization of the prepared pSiNs reveals the nanowires are single crystalline silicon. Please provide which crystal facet they extend along.
4. The authors should add some literature descriptions to make the manuscript more convincing. I would like to suggest the authors cite the following relevant articles:
Centimeter-Long Single-Crystalline Si Nanowires, Nano Letters 2017 17 (12), 7323-7329.

REVIEWER COMMENTS

Reviewer #1 (Remarks to the Author):

My concerns with this manuscript are a) the lack of or at least confusing explanation on whether we are dealing here with lithium silicide or silicate or a mix,

We agree that the silicide-silicate discussion was unclear. We have now clarified the discussion of the chemical analysis and added additional core-level XPS studies to elucidate these points.

and b) the lack of comparison with lithium bioceramics already described in the literature for the same application.

It is indeed valuable to compare our results to lithium-incorporating bioceramics. We now include an additional in vivo study comparing LipSi to lithium-substituted bioglass.

My detailed comments are as follows:

- The introduction is nice and it suggests that Li_xSi_y materials used in lithium batteries are flammable and reactive. The lithium incorporation approach used here probably produces lithium silicate or lithium in ionic form which of course is not flammable. That indicates that this critical difference is not fully appreciated by the authors.

We appreciate the difference between lithium silicide (Li_xSi_y), lithium silicate and ionic lithium species, their different reactivity and solubility profiles. We discussed lithium batteries to highlight the capability to incorporate large and tuneable amounts of lithium within silicon and its broad range of applicability, while highlighting limitations that prevent their application in biology. Indeed the differences between our process and pre-lithiation strategies most likely yields lithium silicates and ionic lithium although we cannot fully exclude the presence of other species within the bulk of the silicon structure.

As the reviewer highlighted that the structure of current discussion can be confusing to the reader, we have revised that section of the introduction to better reflect our argument.

“While pre-lithiation allows incorporating large and controllable amounts of lithium within porous silicon, it typically yields pyrophoric and highly reactive materials³⁵. It would be appealing to develop similarly effective lithiation approaches suitable for biological applications.”

- That leads me to the results part where the authors put a lot of effort into characterising their materials. But they do seem to draw inconsistent conclusions. In some parts they talk about lithiation indicating the formation of lithium silicide (XPS, EELS) and lithium silicate in other parts (Raman, NMR-MAS, XANES). This would need to be teased out by further studies including using high resolution XPS.

We have now revised the analysis of the material composition to improve clarity for the reader. We have furthermore included additional high resolution core level XPS results that confirm the formation of surface silicates (Figure S11). The discussion in the manuscript has now been simplified by only discussing the direct evidence for the presence of lithium silicates in crystalline and amorphous form within the material while mentions of lithium silicides have been removed. Several sentences have been altered in the chemical characterisation section of the manuscript to reflect this.

- The material should also characterised before and after heat treatment at 450 deg C.

The original manuscript already included SEM, HRTEM, ICP-MS, Raman and XRD characterisation of the porous silicon material prior to heat treatment in comparison to heat-treated LipSi samples. We have now additionally characterised the LiCl+pSi physical mixture by XPS prior to heat treatment in comparison to LipSiNs (Figure S11a). The results indicate that the LiCl+pSi mixture is characterized by lithium with higher binding energy which incorporates into the silicon-oxide structure upon heat treatment, leading to the lowering of the binding energy in LipSi. The findings are discussed in the manuscript.

“X-ray photoelectron spectroscopy (XPS) analysis of the core level spectra of Si 2p and Li 1s before and after thermal treatment at 450 °C confirms the incorporation of lithium within the silicon nanowires (Fig S8). The Si 2p spectra show increasing oxidation between the lithium and silicon mixture and the treated LipSiNs, as the peak at 103.8 eV assigned to SiO₂ becomes more prominent after heat treatment, especially in air. Suboxides are also present at lower binding energies between the bulk oxide and the Si₀ doublet, which may contain contribution from the lithium silicate structures present in LipSiNs. Similarly, the binding energy of the Li 1s peak is reduced from the initial 57.2 eV of LiCl to 55.8 – 56.2 eV, which can be attributed to lithium oxides forming following heat treatment.”

- The absence of chlorine signal in XPS (from LiCl) could still mean that lithium ions are adsorbed to the oxidised silicon surface.

The XPS results do not support the hypothesis of a significant adsorption of ionic lithium on the surface while suggesting chemical bonding of the lithium to the silicon nanowires. Furthermore, EELS analysis indicate that the amount of lithium present increases with the thickness of the material, suggesting that lithium loading at the surface provides a minimal contribution in comparison to lithium doping of the porous silicon particle. We have revised the manuscript to better reflect this discussion.

“Chlorine could not be detected by XPS suggesting that the contribution to lithium content from residual LiCl within the mesoporous structure is minimal(Figure S10). The reduced Li binding energy in LipSi compared to physical mixture of pSi and LiCl further support that the lithium on the surface of LipSiNs is preferentially coordinates with the silicon in the nanowire forming lithium silicate structures with varying degrees of oxidation (Figure S8).”

- On page line 184 the authors talk about Si ions being released. I assume they mean silicate ions.

pSi is known to dissolve in orthosilicic acid form and the lithium silicates will also contribute silicate ion release (ref no. 25 in manuscript, doi:10.1002/pssa.200306519). We have revised the manuscript and included a reference to the mechanisms of porous silicon dissolution.

“When exposed to phosphate buffer, LipSiNs do not exhibit visible reactions and dissolve over time analogously to pSiNs, releasing Li and silicate ions including orthosilicic acid²⁵”

- The influence of the heat treatment on lithium ion release should also be presented.

LipSi 5% and LipSi 4% were processed at 450 °C and 650 °C respectively. The higher processing temperature LipSi 4% yielded a material with higher crystallinity than LipSi 5% which displayed a large amorphous shell. The lower solubility of the crystalline forms present within LipSi 4% likely contributed to the slower lithium release observed from the material, compared to the more rapid release likely contributed by the amorphous species present in LipSi 5%. This discussion is now presented in the manuscript.

“The processing temperature influences LipSiNs crystallinity and in turn regulates the release profile. Indeed, in the high-temperature LipSi-4% (650 °C) the crystalline phase with low-solubility extends all the way to the surface preventing the initial rapid lithium release and slowing silicon release compared to pSi. The Li-rich amorphous shell of the low-temperature LipSi-5% (450 °C) and LipSi-1.2% (450 °C) instead contain species that dissolve rapidly contributing to a burst Li release; once the crystalline core is exposed a sustained release occurs analogous to LipSi-4%. The XRD data and the low processing temperature suggest that LipSi-1.3% (450 °C) also possess an amorphous shell and a crystalline core, contributing to its observed dual-release analogous to LipSi-5% and LipSi-1.2%.”

- As for the in vivo model and the overall novelty of the work, it would be important to study any significant differences in terms of the performance compared to other lithium doped bioceramics (including silicon based ones like MSN) which have already been studied for the same purpose. The benchmarking against GTR is useful but not sufficient in my opinion.

We have now incorporated an additional animal study comparing lithium-substituted bioglass to LipSi (figures S20-23). This study shows that LipSi outperforms lithium-substituted bioglass for bone mineral density and bone volume at 2 weeks and 6 weeks. The manuscript has been revised to discuss this data.

“We also compared the regenerative capability of LipSiN to 100% lithium-substituted bioglass (Li-BG), as a model of lithium and silicate ion with an established stimulatory capacity (Figure S21-23). MicroCT analysis showed that both Li-BG and LipSiN stimulated the regeneration of the bone defect, with LipSiN outperforming Li-BG for both bone volume and bone mineral density at 2 wks and 6 wks (Figure S21). Histological analysis revealed that LipSi and Li-BG stimulated the regeneration of new cementum and soft tissue (Figure S22) and immunohistochemistry showed the formation of comparably vascularised tissue for LipSi and Li-BG (Figure S23). Overall LipSiN indicated a comparable effectiveness to Li-BG with a superior capability for bone regeneration.”

- Is there a good justification of using porous silicon nanowires which are more difficult and expensive to produce at scale?

Metal assisted chemical etching (MACE) is a low-cost, room-temperature, solution based, scalable method to produce porous silicon nanomaterials, including nanowires and nanoparticles (ref no 37, doi:10.1002/adfm.201000360). MACE can use inexpensive silicon feedstock, including metallurgical grade silicon, and silicon from biosources (refs no 48, 49, doi:10.1149/1.3548513, doi:10.1007/s12633-012-9129-8). The chemicals employed for production are likewise inexpensive and broadly available.

Overall, the broad tunability for lithium incorporation and release kinetics of the material, combined with its affordable and scalable manufacturing approach, make the resulting material a viable choice for industrial-scale manufacturing. We now discuss this aspect in the manuscript.

“Porous silicon nanowires produced by metal assisted chemical etching are a compelling material for biomedical applications. Conventional silicon nanowire manufacturing processes, such as vapor-liquid-solid growth, require high-temperature processes, highly controlled environments and high purity starting materials, leading to limited scalability and high costs. Metal assisted chemical etching instead is a room-temperature, solution-based process that can use low grade and biosourced silicon to yield large quantities of porous silicon at competitive costs.”

Reviewer #2 (Remarks to the Author):

The paper by Kaasalainen et al. developed lithiated porous silicon nanowires for periodontal regeneration. Authors demonstrated the fabrication process and its characterization using a variety of techniques to support their conclusions. Overall this work is very appealing for NC and deserves consideration.

Considering the application of these nanowires for biomedical purposes, I'd recommend some more additional experiments before publication.

1. Please provide the release studies in Fig. 5a also in simulated oral/salivary fluids. This better represents the biomedical application.

We have conducted the additional release studies in simulated salivary fluid (Figure S15). The release of lithium from LipSi 1.2% and 5% follows the same dual-kinetic pattern of phosphate buffer with an early burst release, attributed to the amorphous layer, followed by a sustained release, attributed to the crystalline phase. Silicon release is slowed as expected by the lower pH of the salivary fluid (pH 4.9) compared to phosphate buffer (pH 7.4). The hydrolysis of porous silicon is known to be highly pH dependent and driven by OH⁻ concentration (ref no. 26, doi:10.1002/pssa.200306519). These results are discussed in the manuscript.

“When LipSi 1.2% and 5% are exposed to simulated salivary fluid to mimic the oral environment, LipSiNs retain their tunable dissolution, displaying a burst early Li release from the amorphous layer, followed by a sustained release from the crystalline structure (Figure S15). Silicon release rate is slowed with respect to phosphate buffer, due to the reduced hydrolytic process in the lower pH environment.”

2. The release of both ions is from a few days to several hours. Authors should demonstrate that the amounts in such times have a therapeutic effect or enough to lead to a therapeutic outcome.

The in vitro data provided in the original manuscript generated using conditioned medium (Figure 5) supported that the amount of ions released have a therapeutic effect. We have performed additional in vitro studies with cells directly in contact with LipSiNs, rather than pre-conditioned medium (Figure S18). These cells experience a real-time exposure to the released ions, to strengthen the evidence regarding the efficacy of the release kinetics. The data shows that LipSi upregulates Axin-2 expression after 24h and the osteogenic genes Runx2 and BMP2 after 3 and 14 days. This additional data is now discussed in the manuscript.

“LipSiN in direct contact with cells also stimulate Wnt/ β cat and osteogenesis (Figure S18). Incubation with LipSiNs reduces cell proliferation above 50 μ g/ μ l concentration, while upregulating Axin2 expression and stimulating osteogenesis at concentrations above 10 μ g/ μ l. In this setup, cells are directly exposed to LipSiN dissolving for up to 14 days, indicating that the ion release profile over the relevant timeframes has a stimulatory effect towards the Wnt/ β cat pathway and osteogenesis over the desired timeframe.”

3. Authors have to demonstrate that there are no adverse effects in case the nanowires end-up elsewhere besides the oral cavity.

We now include histological data from the liver and kidney of control and LipSi-treated animals which does not show evidence of toxicity (figure S20). The data is discussed in the manuscript.

“Histological analysis of the Liver and Kidney of LipSi-treated animals was comparable to the control group, confirming the lack of toxicity to organs beyond the oral cavity (Figure S20)”

4. Authors have to demonstrate by using a control of a mixture of equal amounts of Si and Li found in the nanowires also as an hydrogel formulation to clear prove the nanowire is needed for the intended application.

A large body of existing research indicates that formulating materials incorporating lithium and silicon for the tailored release of bioactive ions has beneficial impact on regenerative capacity. These include for example the body of literature on bioceramics such as lithium-substituted bioglasses (Cordero et al. doi:10.1016/j.jdent.2020.103575) and Laponite® clays (Tomas et al. doi: 10.1016/j.nano.2017.04.016) that discuss the central role of the ion leaching process in the regenerative properties of the resulting materials. As such the concept of developing materials releasing lithium and silicide ions for regeneration is an established principle, that has been successfully exploited extensively.

Our original study already included reference groups for lithium chloride and porous silicon which showed negligible regenerative effect (Figure 8). We now additionally include *in vivo* data from lithium-substituted bioglass which displays a lower regenerative capacity than LipSiN (Figures S21-23), which further support our argument for the development LipSiN as an effective regenerative material for the periodontum.

Thus in accordance with 3R principles for reducing the use of animals in research we opted not to include a group for Lithium + Silicon ions to the new *in vivo* study as it would not provide sufficient significant additional evidence to the efficacy of our approach to justify animal use.

5. In addition, a control using a commercial available product for this medical application should be included in the study to demonstrate the benefit of this novel approach compared to existent ones.

We agree on the usefulness of comparing LipSi with other materials that provide lithium and silicon ions. However we could not identify commercially available products that we could source. As such we sourced 100% lithium-substituted bioglass from a collaborator and compared its use to LipSi in a new animal study. The study showed LipSi producing more bone volume and bone with higher mineral density than the bioglass at 2 week and 6 weeks (Figures S21-23). The data is discussed in the manuscript.

“We also compared the regenerative capability of LipSiN to 100% lithium-substituted bioglass (Li-BG), as a model of lithium and silicate ion with an established stimulatory capacity (Figure S21-23). MicroCT analysis showed that both Li-BG and LipSiN stimulated the regeneration of the bone defect, with LipSiN outperforming Li-BG for both bone volume and bone mineral density at 2 wks and 6 wks (Figure S21). Histological analysis revealed that LipSi and Li-BG

stimulated the regeneration of new cementum and soft tissue (Figure S22) and immunohistochemistry showed the formation of comparably vascularised tissue for LipSi and Li-BG (Figure S23). Overall LipSiN indicated a comparable effectiveness to Li-BG with a superior capability for bone regeneration.”

Reviewer #3 (Remarks to the Author):

In this manuscript, the authors reported a kind of “lithiated porous silicon nanowires (LipSiNs)” strategy to generate a biocompatible and bioresorbable material for treating Periodontal disease. Prelithiation, a strategy developed for battery technology, is proposed via the technology to stimulate bone repair via silicic acid release while providing regenerative stimuli through lithium activation of the Wnt/ β -catenin pathway. Therefore, the unique structure exhibited a good treatment method. However, there are still some problems as follows before published:

1. To date, SiNWs were fabricated by vapor-liquid-solid (VLS) process, which has not been possible to prepare in batches. Please compare with the preparation method of LipSiNs, including cost, preparation process, and yield, and explain the advantages of the work.

Metal assisted chemical etching (MACE) is a low-cost method to produce porous silicon nanomaterials, including nanowires and nanoparticles. MACE can use inexpensive silicon feedstock, including metallurgical grade silicon, and silicon from biosources. The chemicals employed for production are likewise inexpensive and broadly available. Since the process is solution-based and can start from nanoparticle feedstock, it can be easily scaled.

Overall, the broad tunability for lithium incorporation and release kinetics of the material, combined with its affordable and scalable manufacturing approach, make the resulting material a viable choice for industrial-scale manufacturing. We now discuss this aspect in the manuscript.

“Porous silicon nanowires produced by metal assisted chemical etching are a compelling material for biomedical applications. Conventional silicon nanowire manufacturing processes, such as vapor-liquid-solid growth, require high-temperature processes, highly controlled environments and high purity starting materials, leading to limited scalability and high costs. Metal assisted chemical etching instead is a room-temperature, solution-based process that can use low grade and biosourced silicon to yield large quantities of porous silicon at competitive costs.”

2. The size distribution of LipSiNs should be provided, including the length and diameter of nanowires. This data is now reported in Figure S6 and discussed in the manuscript

We have included a scanning electron microscopy quantification of the length and diameter of the nanowires (Figure S6) and discussed it in the manuscript.

“Lithiation preserves the shape of nanowires (Figure 2a,b) which retain their length and diameter (Figure S6).”

3. HRTEM characterization of the prepared pSiNs reveals the nanowires are single crystalline silicon. Please provide which crystal facet they extend along.

Nanowires are produced from $\langle 100 \rangle$ single-crystal silicon wafers, extending orthogonally to the surface in the $\langle 100 \rangle$ direction. We now include this information in the manuscript and in the caption to Figure 3a.

“High resolution transmission electron microscopy (HRTEM) of LipSi-5% show the nanowires extending along the $\langle 100 \rangle$ direction with an inner core single-crystal structure surrounded by an amorphous shell (Figure 3a).”

4. The authors should add some literature descriptions to make the manuscript more convincing. I would like to suggest the authors cite the following relevant articles: Centimeter-Long Single-Crystalline Si Nanowires, Nano Letters 2017 17 (12), 7323-7329.

We have now incorporated the suggested reference as Ref no. 25.

Reviewers' Comments:

Reviewer #1:

Remarks to the Author:

The authors have carefully addressed my comments and I hence recommend accepting this manuscript.

Reviewer #2:

Remarks to the Author:

Authors have revised extensively the paper according to previous suggestions. I recommend acceptance of the paper as it is now.

Reviewer #3:

Remarks to the Author:

The authors have addressed the comments/suggestions carefully. I am satisfied with the revisions. This manuscript is now acceptable for publication.